# QUANTIFYING THE STRUCTURAL GAP: HIERARCHICAL PRIORS STABILIZE RECURSIVE LEARNING IN TRANSFORMERS

## ABSTRACT

Standard Transformers can theoretically represent recursive structures, but they often exhibit significant data inefficiency when learning from hierarchical sequences. We investigate this "structural gap" using Context-Free Grammars (CFGs) as a controlled experimental environment. Our analysis reveals a sharp phase transition: Transformer performance collapses as the depth of the latent hierarchy increases, regardless of parameter count. To understand this bottleneck, we introduce the Hierarchical Transformer (HT) as a structural intervention. HT employs a ParentEmbedder to organize computation across multiple levels while maintaining strict autoregression. We demonstrate that HT reaches full structural correctness with 50% less data than Flat Transformer baselines (FT) and effectively shifts the phase transition boundary for deep grammars. Finally, we show that these mechanistic benefits transfer to real-world multi-scale data, represented in this work by discretized protein sequences. Our results suggest that the failure of flat attention on recursive data is primarily an optimization bottleneck that can be resolved by aligning model topology with the latent hierarchy of the task.

## 1 INTRODUCTION

While Transformers serve as the backbone for state-of-the-art AI, it is surprising how these models can struggle with relatively simple hierarchical tasks. Despite being universal approximators (Hahn, 2019), they often lack the native inductive biases required to resolve multi-scale organization efficiently. This leads to a sharp **phase transition** in structural learning: performance remains stable for moderate hierarchies but collapses as depth increases (Allen-Zhu & Li, 2024). We treat Context-Free Grammars (CFGs) as a testbed to isolate this bottleneck, following methodologies in mechanistic interpretability (Nanda et al., 2023; Michaud et al., 2023).

Our work makes the following contributions to understanding hierarchical learning:

- **Characterization of the Structural Gap:** We quantify the data inefficiency of flat Transformers and relate this gap to the "Quantization Hypothesis" (Michaud et al., 2023).

- **Structural Intervention via HT:** We propose the Hierarchical Transformer (HT), which externalizes the parsing process into discrete levels via a *ParentEmbedder* while maintaining strict autoregression.

- **Evidence of Optimization Stability:** We demonstrate that HT achieves structural correctness using half as many training samples as flat models, suggesting that explicit hierarchy simplifies the optimization landscape.

- **Validation on Physical Sequences:** We verify these findings on discretized protein sequences (ATOM3D Townshend et al. (2021)). We show that when model topology mirrors physical hierarchy (atoms $\rightarrow$ residues), optimization becomes significantly more stable.

## 2 RELATED WORK

**Diagnostic Tools & CFGs.** CFGs are the standard formalism for recursion in language (Chomsky, 1957). Recent work uses them to probe neural limits, showing that while Transformers can identify nonterminal boundaries, they struggle with depth scaling (Allen-Zhu & Li, 2024).

## 3 PRELIMINARIES: CONTEXT-FREE GRAMMARS

To study how models learn recursive structures, we use Context-Free Grammars (CFGs) as a controlled experimental setup. A CFG is defined by a set of production rules that dictate the expansion of a single non-terminal symbol (a parent) into a sequence of child symbols. We give below a simple example, and refer to Allen-Zhu & Li (2024) for a more formal definition.

**Example: A Two-Level Grammar**  Consider a grammar with a root symbol $S$ (Level 0), intermediate non-terminals $\{\alpha, \beta, \gamma\}$ (Level 1), and terminal leaves $\{1, 2, 3, 4, 5\}$ (Level 2). The production rules are defined as follows:

- **Level 0 $\rightarrow$ 1:** $S \rightarrow (\alpha, \beta) \mid (\alpha, \gamma) \mid (\gamma, \gamma)$
- **Level 1 $\rightarrow$ 2:**
  - $\alpha \rightarrow (2, 3) \mid (5, 1)$
  - $\beta \rightarrow (3, 3) \mid (2, 4) \mid (1, 1)$
  - $\gamma \rightarrow (3, 5) \mid (1, 2)$

To generate a sentence, we select rules uniformly at random to expand each symbol. For instance, a sequence can be derived through the following expansion:

$$S \xrightarrow{\text{Level 1}} (\alpha, \gamma) \xrightarrow{\text{Level 2}} (5, 1, 1, 2)$$

In this case, the second rule was selected to expand $S$, the first for $\alpha$, and the second for $\gamma$. This mechanic is easily extended to $N$ levels, where each application of a rule increases the depth of the latent tree until only terminal leaves remain. We define the branching factor $c$ as the number of children per parent, equal to 2 at all levels here.

**Aligning Model Topology**  Standard Transformers are "flat" architectures; in the sense that they operate directly on the leaf sequence, $(5, 1, 1, 2)$ in the example above, and must infer latent groupings through self-attention. We propose to *align model topology* by mirroring the grammar's hierarchical structure directly in the architecture. The Hierarchical Transformer (HT) explicitly synthesizes child tokens into parent representations level-by-level through a **ParentEmbedder**.

## 4 METHODOLOGY: THE HIERARCHICAL TRANSFORMER (HT)

We introduce HT, which explicitly encodes the hierarchical structure into the architecture. Unlike flat Transformers (FT) that apply self-attention uniformly across sequences, HT processes the hierarchy level-by-level through a **ParentEmbedder**.

**ParentEmbedder.**  Given leaf embeddings $X \in \mathbb{R}^{T \times d}$ and a chunk size $c$, the ParentEmbedder groups adjacent tokens. Each parent token $P_t$ represents a local structural unit:

$$P_t = \text{ReLU}\left(W_p[X_{tc-c+1}; \dots; X_{tc}] + b_p\right)$$

where $W_p \in \mathbb{R}^{d \times (cd)}$. This dimensionality reduction allows higher layers to attend over a compressed sequence of length $T/c$, effectively increasing the receptive field without expanding the attention window (see Figure 1, up).

**Autoregressive Fusion.**  To maintain the generative property, we implement a **causal structural mask**. Parent features $Z$ are broadcast back to token resolution $T$. For a token at position $i$, the model attends only to parents of strictly preceding blocks. The parent containing token $i$ and all future parents are masked. This ensures the prediction of $x_{i+1}$ depends only on $\{x_1, \dots, x_i\}$ and the coarse-grained summaries of completed previous chunks (see Figure 1, down).

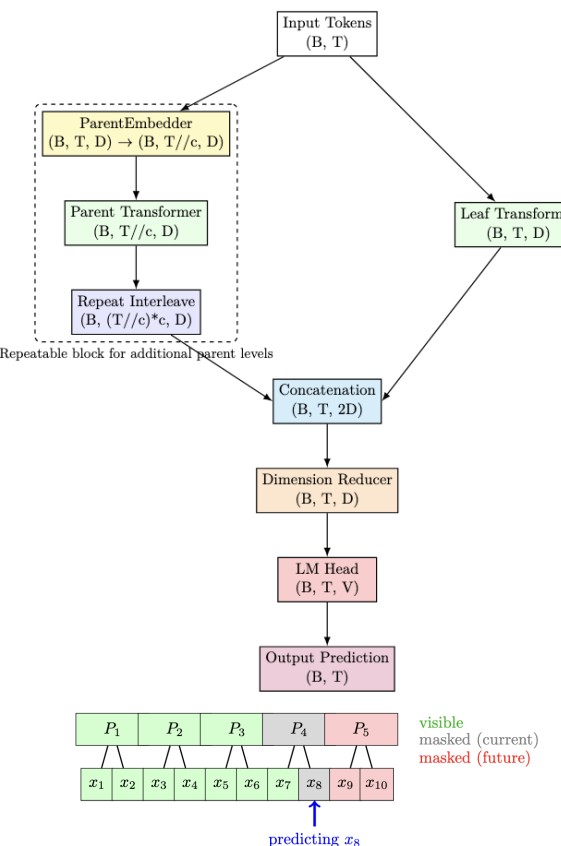

Figure 1: HT architecture. (Up) The ParentEmbedder reduces resolution by composing $c$ children into a single parent token. (Down) The causal structural mask ensures $x_{i+1}$ is predicted using only past tokens and completed parent summaries, preventing data leakage.

## 5 EXPERIMENTAL SETUP: THE CFG LABORATORY

We treat Context-Free Grammars (CFGs) as a controlled environment to isolate structural learning bottlenecks. We generate sequences level-by-level using recursive production rules with fixed branching factors, varying the grammar depth $D \in \{5, \ldots, 9\}$. This setup allows us to precisely quantify how architectural priors affect the learning of latent hierarchies as they scale in complexity.

**Evaluation Metric.** We assess structural mastery via generative sampling. After each training epoch, the models generate 100 sequences, which we verify against the ground-truth CFG production rules. We define the **Samples to Structural Mastery** (our measure of sample efficiency) as the number of training samples seen at the first instance where 100% of generated sequences are valid.

Table 1: CFG Structural Parameters and Sample Efficiency. We compare branching factor and depth against the number of training samples required to reach Structural Mastery.

| CFG ID | Depth | Branching (Top → Bottom) | FT Samples | HT (Ours) |
|--------|-------|--------------------------|------------|-----------|
| CFG-5L | 5 | [2, 2, 8, 4, 4] | 120,000 | **60,000** |
| CFG-7L | 7 | [2, 2, 2, 2, 2, 4, 4] | 140,000 | **65,000** |
| CFG-9L | 9 | [1, 2, 2, 2, 2, 2, 2, 2, 4] | *Collapse* | **75,000** |

# 6 RESULTS AND ANALYSIS

**Phase Transition and Baseline Collapse.**   Following the experimental framework of Allen-Zhu & Li (2024), we evaluate the scaling limits of flat Transformers as grammar depth increases. While standard Transformers can learn internal mechanisms for structural parsing, we identify a sharp **depth–accuracy phase transition** where this capability breaks down. As shown in Figure 2a, FT performance remains stable for moderate hierarchies but collapses sharply beyond depth 7. Table 1 confirms this "Structural Gap": at depth 9, the flat model fails to reach mastery within the training budget, signifying a fundamental optimization wall in flat attention's ability to resolve deep recursion.

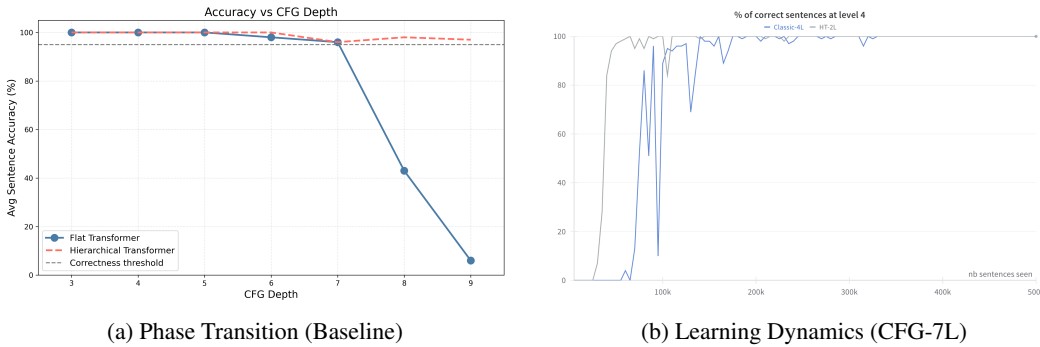

(a) Phase Transition (Baseline)                    (b) Learning Dynamics (CFG-7L)

Figure 2: (a) Accuracy collapses as depth increases for flat models. (b) HT achieves Structural Mastery with $2\times$ less training samples and more stably than FT.

**Sample Efficiency.**   The introduction of hierarchical priors through HT yields a consistent $2\times$ reduction in sample complexity across all solvable grammars. On CFG-7L, HT requires 65,000 samples for structural mastery vs. 140,000 for FT. Crucially, HT shifts the transition boundary, resolving the depth-9 grammar where the baseline collapses (Table 1).

**Transfer to Physical Sequences.**   To check if these benefits transfer to other domains, we evaluate HT on discretized protein sequences (ATOM3D dataset). By aligning the architecture with the atom→residue hierarchy, HT achieves lower Kabsch-aligned MSE and faster convergence than flat baselines, confirming that structural alignment is effective for real-world multi-scale data.

# 7 DISCUSSION: MECHANISTIC INSIGHTS

Our results suggest the "structural gap" is caused by a **coupling of errors** (Dehghani et al., 2019): flat models must learn segmentation of CFG sentences into words and rules simultaneously. HT provides:

1. **Path Length Reduction:** Operating on coarse tokens reduces attention steps for long-range constraints, mitigating "over-squashing". This multi-scale propagation is essential for resolving dependencies in complex domains (Dwivedi et al., 2023).
2. **Gradient Scaffolding:** Explicit parent-child mappings via the *ParentEmbedder* ensure structural signals for deep grammars are preserved. Gradients for local structures (e.g., atoms) propagate to higher-level reasoning (e.g., residues) without interference from global noise.

# 8 CONCLUSION

By using CFGs as a scientific testbed, we have quantified the inefficiency of flat Transformers in learning deep hierarchies. Our intervention with the Hierarchical Transformer demonstrates that architectural alignment with latent structure is a key factor in resolving optimization bottlenecks in recursive data.

**Reproducibility Statement** All code, grammar specifications, and training configurations needed to reproduce the experiments are provided in the supplementary material and will be released in an open-source repository upon publication.

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
