# OpenReview forum: "Quantifying the Structural Gap: Hierarchical Priors Stabilize Recursive Learning in Transformers"
_ICLR.cc/2026/Workshop/Sci4DL — Submitted to Sci4DL 2026_

### Official Review · Reviewer_dt9H · 2026-02-11

**Fit:** 2
**Significance:** 2
**Confidence:** 2

**Summary:**

The authors introduce Hierarchical Transformers (HTs), which adapt transformers by explicitly encoding a hierarchical structure. This is done by adding a parent embedder that aggregates the information from subsequences of n tokens. The parent embeddings result in a shorter sequence that also undergoes self-attention, and the outputs get concatenated to those of the standard self-attention over the original token space.

**Strengths:**

Overall the writing is relatively clear and the core idea is intuitive to understand. There are some interesting experimental findings, showing that flat transfoermers learn poorly for deep DFGs, but the hierarchical transformers learn well. In particular, it seems like the hierarchical transformer method can improve performance on the ATOM3D discretized protein sequence dataset. The study of transformer's expressivity and optimization is an interesting area that is relevant to the workshop's theme.

**Suggestions:**

Suggestions:
1. Many important experimental details are missing, including information on the transformer architecture used, optimization hyperparameters, and numerical results for ATOM3D. These should at least be added to the Appendix, though condensing Figure 1 might also free up space to describe these details.

2. There is lots more room to explore the HT architecture. For example, the parent embeddings can be projected back into the original token space at intermediate layers of the transformer, rather than having this be a completely separate computation path that gets concatenated only at the end.

Questions:
1. For the hierarchical transformer to work, do we need to know the depth and branching factors a priori? What about practical settings where we don't know how to set these, or cases where the production rules in a given level yield sequences of different lengths?

2. Why were those particular branching factors selected? And is a branching factor of 1 (in the first layer of the depth-9 experiments) really meaningful?

---

### Official Review · Reviewer_gzhE · 2026-02-22

**Fit:** 2
**Significance:** 1
**Confidence:** 2

**Summary:**

This paper studies how conventional Transformers might be inefficient on learning PCFGs with deep hierarchical structure, and proposes a hierarchical transformer architecture that improves on this aspect.

**Strengths:**

This paper uses PCFGs as a testbed that provides easy control on data complexity, useful for understanding Transformer performance and limitations. They also attempt to extend their analysis to practical settings for protein sequences (however also see 4. in suggestions).

**Suggestions:**

Suggestions/questions/comments:


1. The hierarchical attention idea is not exactly new, and has been studied before (e.g. [1] and related work discussion therein). Could the authors clarify how their approach is meaningfully different from these prior works?

2. There are no details about depth/width of models used, learning rate, batch size etc. (the Reproducibility Statement mentions supplementary material but I could not find any on OpenReview). I would recommend the authors to add these details in the main paper or appendix.

3. "...relate this gap to the "Quantization Hypothesis"...": I could not find any subsequent discussion or justification of this claim in Section 1, could the authors clarify?

4. “Transfer to Physical Sequences” result simply mentions “lower Kabsch-aligned MSE and faster convergence” without quantifying or elaborating on these claims. I recommend the authors add a discussion on these terms for readers who are not familiar with this research area.

5. Why is the ParentEmbedder $P_t$ non-negative? I’m also not sure what the “Repeatable block for additional parent levels”, or the “Dimension Reducer” block means in Fig 1, and would encourage the authors to discuss this in the main text.

6. Table 1: Why is there a branching factor of 8 in CFG-5L (all others are either 1, 2 or 4)? Further, CFG-9L has a branching factor of 1 on the first level, and I’m not sure if this increases the ‘difficulty’ of the task.


[1] Chalkidis et al., 2022. An Exploration of Hierarchical Attention Transformers for Efficient Long Document Classification. Arxiv: 2210.05529

---

### Official Review · Reviewer_bgXh · 2026-02-24

**Fit:** 1
**Significance:** 1
**Confidence:** 3

**Summary:**

This paper addresses hierarchical learning in a controlled transformer setting trained on CFGs. The authors vary CFG depth and compare the conventional transformer architecture, to a newly proposed architecture that is tailored to the hierarchical setting discussed in the paper, and aligned with the depth of the CFG it is being trained on via a hyperparameter.  The paper has several problems that make it impossible to evaluate the correctness of the claims and the stated contributions in the current form, in particular the lack of details in the experimental setup and the incomplete presentation of results and the subsequent evaluation thereof. Due to this, I suggested that the paper be rejected. See Suggestions for a more thorough explanation of this reasoning.

**Strengths:**

* The chosen problem setting is interesting if well motivated.
* The chosen setting and CFGs are well suited for small controlled experiments.

**Suggestions:**

1. Issues with motivation: The authors cite (Allen-Zhu & Li, 2024) for claiming that conventional transformer "performance remains stable for moderate hierarchies but collapses as depth increase". The paper they cite finds CFG depth differences for encoder-only architectures when comparing to autoregressive models, but I cannot find any mention of "collapse" for bigger depths in autoregressive models in the cited paper. It would be helpful to clarify this issue, or at least present evidence of this failure mode, see next point.
2. Various important details and results left out of the main body, and there is no Appendix, which make it impossible to evaluate the main contributions of the paper:
	* One of the main claims of the paper, is that conventional transformers will "collapse" for deeper hierarchical structures, regardless of parameter count. This is illustrated in Fig. 2a), but no detail or argumentation is provided regarding the parameter count of this model (ie does the model have sufficiently many parameters to learn this CFG depth). Additionally, no results are shown for other parameter counts than the default setup, and it is not clear how these were varied.
	* HT architecture, as shown in Fig. 1, is not fully described. Section 4. only introduces the ParentEmbedder and Autoregressive Fusion, but leaves out many of the other components.
	* Claimed but unsubstantiated results on discretized protein sequences, which should serve as evidence to the generalization of the proposed architecture. It would be helpful to see a plot or a table showcasing these results, alongside model and training details. Additionally, this section would benefit from a more thorough explanation of the discretized protein sequences as an interesting and more general setup.
	* Authors claim that part of these missing elements will be made available upon publication, but these are necessary to evaluate and review the correctness of their claims and results, so the paper is incomplete without them. I suggest that the aforementioned missing points be addressed in subsequent iterations of this work.
3. Mechanistic insights are unsubstantiated. This section provides a potentially a plausible, but previously unaddressed hypothesis that the poor performance of flat transformers is due to coupling errors.. If the points above are all addressed, this hypothesis is an interesting path to pursue to gain mechanistic insights on the nature of the issue, but the hypothesis is not validated by the experiments presented in the paper, as implied in the discussion.

---

### Meta-Review · Area_Chair_GG6v · 2026-03-02

**Recommendation:** Reject

**Metareview:**

The paper studies the role of a hierarchical prior in a transformer for the learnability of a CFG. Although the question and the basic ideas are very interesting, the paper has many unsubstantiated claims and lacks proper reporting of experimental settings.

---

### Decision · Program_Chairs · 2026-03-02

Reject